# Post-Translational S-Nitrosylation of Proteins in Regulating Cardiac Oxidative Stress

**DOI:** 10.3390/antiox9111051

**Published:** 2020-10-28

**Authors:** Xiaomeng Shi, Hongyu Qiu

**Affiliations:** Center for Molecular and Translational Medicine, Institute of Biomedical Science, Georgia State University, Atlanta, GA 30303, USA; xshi8@student.gsu.edu

**Keywords:** S-nitrosylation, nitric oxide synthase, redox, heart, ischemia, cardiac protection

## Abstract

Like other post-translational modifications (PTMs) of proteins, S-nitrosylation has been considered a key regulatory mechanism of multiple cellular functions in many physiological and disease conditions. Emerging evidence has demonstrated that S-nitrosylation plays a crucial role in regulating redox homeostasis in the stressed heart, leading to discoveries in the mechanisms underlying the pathogenesis of heart diseases and cardiac protection. In this review, we summarize recent studies in understanding the molecular and biological basis of S-nitrosylation, including the formation, spatiotemporal specificity, homeostatic regulation, and association with cellular redox status. We also outline the currently available methods that have been applied to detect S-nitrosylation. Additionally, we synopsize the up-to-date studies of S-nitrosylation in various cardiac diseases in humans and animal models, and we discuss its therapeutic potential in cardiac protection. These pieces of information would bring new insights into understanding the role of S-nitrosylation in cardiac pathogenesis and provide novel avenues for developing novel therapeutic strategies for heart diseases.

## 1. Introduction

Physiological nitric oxide (NO) is synthesized by L-arginine through three types of nitric oxide synthases, namely neuronal nitric oxide synthase (nNOS), inducible nitric oxide synthase (iNOS), and endothelial nitric oxide synthase (eNOS) [1]. NO is a free radical with a single, unpaired electron, which makes it highly labile (a half-life of only a few seconds or less) and chemically reactive [2]. Thus, the bioactivity of NO can be partially compromised because NO rapidly reacts with numerous inactivating species present in the bloodstream and cellular milieu [3]. Reduced protein thiol groups were reported decades ago to serve as rich NO carriers that could not only stabilize NO and extend its half-life but also protect its biological activity from oxidative inactivation [4].

The term S-nitrosylation was first coined by Jonathan S. Stamler in 1992 to describe the reversible formation of the S-nitrosothiols (SNOs) of reduced thiols from protein sulfhydryl groups that were exposed to NO [5]. Most proteins in the human body possess such thiol groups and can function as substrates for S-nitrosylation, making it a ubiquitous post-translational modification (PTM) in biology [6]. Thus far, more than 4000 SNO sites involving over 3000 proteins have been experimentally identified and curated in terms of the structural characteristics, functionality, disease relevance, and regulatory networks of S-nitrosylated proteins in dbSNO, the first database of cysteine S-nitrosylation [7]. Since NO functions as an integrative element in electron transfer reactions, S-nitrosylation is now considered as a key mechanism that mediates extensive redox-based cellular signal transduction.

Like other PTMs of proteins, such as phosphorylation, S-nitrosylation has emerged as a key regulatory mechanism in various cellular functions and has been studied in many physiological and disease conditions. It has been reported that S-nitrosylation plays an important role in cellular functions in aspects of gene regulation, immune modulation, vascular homeostasis, and respiratory and neuronal signaling. Conversely, its impairment has been implicated in many diseases such as neurodegenerative diseases, airway diseases, diabetes, and endotoxic/septic shock. In recent years, the significance of protein S-nitrosylation has been revealed in the heart, leading to a great number of discoveries in the mechanisms underlying the pathogenesis of heart diseases. Emerging evidence indicates that S-nitrosylation is a potential target of cardiac protection that would provide new avenues for developing new therapeutic strategies.

In this review, we summarize the recent studies in understanding the molecular and biological basis of S-nitrosylation, including its formation, spatiotemporal specificity, homeostatic regulation by transnitrosylation and denitrosylation, and its association with cellular redox status. We also outline the currently available methods that have been applied to detect SNO. We additionally sum up the up-to-date studies in the implications of S-nitrosylation in cardiovascular diseases and discuss its therapeutic potential in cardiac protection. This information could bring new insights into understanding the role of SNOs in cardiac function and protection against heart diseases.

## 2. Molecular Basis of SNO Formation

S-nitrosylation, by definition, is the covalent attachment of endogenous NO to the thiol (or “sulfhydryl”) side chain of cysteine to form an SNO. Endogenous SNO moieties are much more stable than NO itself, serving as NO donors and participating in NO metabolism [8]. S-nitrosylation can occur not only in proteins to produce SNO proteins but also in low-molecular-weight (LMW) thiols that make low-molecular-weight S-nitrosothiols (SNO-LMW). SNO has a typical spatiotemporal specificity that is highly associated with subcellular redox compartmentalization. Two major consensus motifs of SNO sites, the acid-base motif and the hydrophobic motif, contribute to understanding the target selectivity and specificity of S-nitrosylation. Two main mechanisms regulate the homeostasis of SNO-dynamic transnitrosylation and denitrosylation—to maintain normal cell function.

### 2.1. SNO Protein and SNO-LMW

Cysteine is unique among the coded amino acids of a protein. It is the only one that carries a thiol group, an organic compound containing a sulfhydryl (-SH) functional group with a sulfur atom bonded to a hydrogen atom [9]. This highly reactive thiol side chain places cysteine in a unique position where any other amino acid cannot replace it. Though cysteine is one of the least abundant among the 20 common amino acids in proteins, it often occurs in functionally important sites of proteins [10]. Free cysteines are classified as polar and highly hydrophobic, and they tend to be either poorly conserved or highly conserved. In contrast, cysteines within proteins tend to deviate from the properties of free cysteines in varying degrees to adjust their reactivity under specific protein environments [11]. Only one or a few cysteine residues of a protein are targeted by NO at the physiological level. Still, they are usually sufficient enough to alter the structure, reactivity, stability, and function of the protein [12].

In addition to the SNO in proteins, SNO also occurs in low-molecular-weight (LMW) thiols that produce SNO-LMW, such as S-nitrosocysteine (CysNO), S-nitrosoglutathione (GSNO), and S-nitroso-CoA (SNO-CoA) [13]. Glutathione is the most abundant low molecular-weight thiol found to date and has been most extensively studied as a key element in regulating cellular redox homeostasis [14]. It is composed of three amino acids, which are cysteine, glutamate, and glycine. The S-nitrosylation of glutathione (GSH) produces GSNO, which serves as a stable and mobile NO reservoir and the main endogenous NO donor to other proteins via transnitrosylation [15].

### 2.2. The Mechanisms of SNO Formation

Several mechanisms have been reported to be involved in SNO formation. First, NO is a poor oxidant that rarely reacts spontaneously with thiol residues in the physiological milieu. Therefore, SNO mostly occurs after oxidative reactions that render NO to higher oxides of nitrogen with strong nitrosylating effects, such as nitrogen dioxide (NO_2_), nitrogen trioxide (N_2_O_3_), or peroxynitrite (ONOO^−^) [12]. Secondly, SNO formation is mediated by a thiyl radical recombination pathway in which a thiyl (RS·) radical that is formed via hydrogen abstraction by another radical (X•) that recombines with NO to generate an S-nitrosothiol [16]. Third, transition metals provide a catalytic mechanism of SNO formation. Cysteines have a unique metal-binding ability and display a high affinity toward oxidized transition metal ions such as Fe^2+/3+^, Zn^2+^, and Cu^2+^. Transition metals incorporated within metalloenzymes have different oxidation states and can rapidly transfer electrons to and from the metal, making them efficient catalysts for redox reactions. More specifically, the transition metals can catalyze the one-electron oxidation of NO to nitrosonium (NO^+^), which can nitrosate a cysteine thiol located close to the catalytic center to form a nitrosothiol [17]. SNOs generated from these various pathways can then be transferred to new thiol groups via transnitrosylation. Fourth, dinitrosyl iron complexes (DNICs), which have been reported to serve as the major cellular forms of NO (NO donor), provide an alternative mechanism to promote cellular RSNO formation. DNICs are formed by the reaction of NO with nonheme ions derived mostly from the intracellular chelatable iron pool (CIP). DNICs have been proposed to mediate the majority of cellular RSNO formation via O_2_-independent transnitrosylation [18]. In 2011, Tsou et al. demonstrated the transformation process of monothiolate-containing DNICs into RSNOs [19].

### 2.3. The Spatiotemporal Specificity of SNO with Redox Compartmentalization

Like other signaling pathways mediated by NO, SNO has a typical spatiotemporal specificity, and its pKa-dependent thiol reactivity has been reported to be one of the major determinants to interpret this specificity [20]. Another key determinant is the colocalization of SNO substrates with NOS within the same subcellular compartments, promoting high local concentrations of reactive nitrogen species (RNS), thus more precisely targeting the redox-sensitive cysteine thiols and facilitating signal propagation [21]. Such a phenomenon runs against the notion that NO exerts its biological activities by simple diffusion down its concentration gradient across cell membranes [22]. Additionally, different subcellular compartments exhibit differential electrochemical characteristics regarding redox potentials, among which mitochondria are considered to comprise the most redox-active compartment with the highest rates of electron transfer and oxidation. Reciprocally, other intracellular compartments, such as the nuclei, are less reducing and relatively resistant to oxidation, followed by the more oxidizing secretary pathway. Cytosol maintains a rather reducing environment while the extracellular compartments are kept at stable oxidizing potentials. The highly intricate redox control networks within each compartment where the catalytic oxidation of substrates vary greatly in favor of or against SNO formation serve as other potential mechanisms for specificity in S-nitrosylation [23]. Moreover, the ubiquity of distinct, non-equilibrium potentials of redox couples within compartments and redox communication between compartments reminds one of the importance of redox regulation in protein function via S-nitrosylation at the organellular level. Redox compartmentalization is a typical example indicating that S-nitrosylation is mainly a short-range mechanism of NO signaling [24].

The discovery of the two main consensus motifs surrounding cysteine residues, the acid-base motif and the ‘hydrophobic compartment’ motif, introduced another level of target specificity of S-nitrosylation. The acid-base motif model was first proposed in 1997 by Stamler et al. [25]. They previously observed that the specific positioning of Cysβ93 between a basic histidine and an acidic aspartate (His-Cys93-Asp) could effectively promote the S-nitrosylation of the highly conserved Cysβ93 residue in hemoglobin. This pattern was detected in many proteins from multiple protein databases, which confirmed the generality of acid-base catalysis in S-nitrosylation. The conformational change was thought to promote the transition from the low to high affinity of thiol-containing proteins for NO during S-nitrosylation. In 2001, Hess et al. proposed a ‘hydrophobic compartment’ motif as an alternative mechanism to elucidate the specificity of S-nitrosylation for proteins that lack the acid-base motif [26]. They reported that the S-nitrosylation of the only thiol of each RyR1 (ryanodine receptor 1) occurred only in the highly hydrophobic, calmodulin-binding domains (CaMBD) without having any physical interaction with the charged amino acids. The high local hydrophobicity within proteins promotes the micellar catalysis of S-nitrosylation on resident Cys residues for which both NO and O_2_ exhibit higher solubility in hydrophobic environments in which they tend to accumulate and concentrate.

All the determinants of SNO specificity mentioned above have been considered to help explain SNO specificity. In 2010, Marino et al. conducted a comprehensive analysis of 70 NO-Cys sites in 55 proteins regarding the general features of SNO that were reported in previous literature [11]. The proteins selected for their dataset were non-redundant proteins with well-characterized three-dimensional protein structures determined by NMR spectroscopy or X-crystallography, and their NO-Cys sites could be experimentally proven in vivo without applying high levels of NO-sylating agents to avoid mistaking experimental artifacts as natural NO-Cys sites. They tested several key parameters reported to be responsible for the specificity of protein S-nitrosylation, such as thiol pKa, S atom exposure, the acid-base motif, and the hydrophobic motif. Their research results indicated that none of the above parameters or any of their combinations exhibited sufficient generality to represent common features of all or even a majority of NO-Cys sites. Thus, they should not be considered for the reliable identification and prediction of SNOs. It is worth pointing out that only 26% of NO-Cys in their dataset had flanking acid-base motifs within a distance of 6 Å of NO-Cys, while 90% of NO-Cys sites had both acidic and basic residues within 8 Å from NO-Cys. This observation indicated a more distant positioning of charged residues regarding the modifiable Cys sulfur atoms. Therefore, they proposed a revised acid-base motif with the presence of oppositely charged amino acids within 6 and 8 Å of NO-Cys, respectively. However, this modified motif did not qualify as a characteristic feature for direct S-nitrosylation, either. Instead, the long-range electrostatic interactions in protein-protein association between the positively and negatively charged amino acids have been thought to serve as an alternative mechanism for elucidating the specificity and selectivity of S-nitrosylation. Therefore, it is postulated that the acid-base motif is likely to function indirectly via protein-protein interactions other than the direct activation of Cys through acid-base catalysis. In a word, the characteristics conferring the specificity of S-nitrosylation are still under intense investigation, and many questions remain largely controversial and unresolved.

### 2.4. Regulation of SNO Homeostasis by Transnitrosylation and Denitrosylation

The cellular homeostasis of SNO is regulated by two main mechanisms: transnitrosylation and denitrosylation. Transnitrosylation is the reversible transfer of an NO group from one cysteine residue to another, thereby allowing for the effective propagation of SNO-based signals [27]. A recent study showed that transnitrosylation could not only occur between low-molecular-weight SNOs and protein thiols but also between two proteins (protein-protein transnitrosylation) [28]. For example, GSNO has been characterized as a powerful transnitrosylating agent that can deliver NO groups to SNO sites remote from NO sources, leading to changes in the activity and function of the far-reaching target proteins [29]. However, not all thiols are equally susceptible to transnitrosylation. The selectivity of transnitrosylation depends on the specific three-dimensional environments of targeted thiol groups, involving both kinetic and thermodynamic factors, particularly thiol steric hindrance and solvent exposure [30].

In contrast, denitrosylation is the reverse reaction of S-nitrosylation via the removal of an SNO moiety from either SNO-proteins or SNO-LMW. From another perspective, transnitrosylation can be considered a denitrosylation process of the donor protein.

S-nitrosylation is generally a nonenzymatic reaction. However, there is accumulating evidence pointing to enzymatic regulations of S-nitrosylation via both S-nitrosylases and denitrosylases, which cooperatively control steady-state levels of SNO [31].

S-nitrosylases: S-nitrosylases can enzymatically mediate the transfer of an NO moiety from either Metal-to-Cys or Cys-to-Cys. Nitrosylase metalloproteins, such as hemoglobin and cytochrome c, can transfer NO groups from redox-active transition metals to cysteine thiol (Metal-to-Cys transfer) via the metal-catalyzed reaction. For example, in mammalian hemoglobin, oxidized hemes (Fe(III)) catalyzes auto-S-nitrosylation of Cys93 (CysβSNO) in a hemoglobin (Hb) β-chain via the intramolecular transfer of NO from iron-nitrosyl hemoglobin (HbFeNO) to Cys93. SNO-Hb also exhibits Cys-to-Cys transnitrosylase activity, which transfers NO to an N-terminal erythrocytic membrane protein, AE1 (anion exchange protein 1) [32]. Cytochrome c heme contributes to a robust formation of GSNO. Ferric cytochrome c (CytFe^Ⅲ^) initially binds weakly to GSH (CytFe^Ⅲ^-GSH). Then ferric cytochrome c is synergically reduced by NO and GSH to cytochrome c (CytFe^Ⅱ^), resulting in GSNO formation [33].

Cys-to-Cys nitrosylases are SNO-proteins that transfer their NO groups to acceptor cysteines on the target protein (Cys-to-Cys transfer) [34]. Alongside hemoglobin, only a few S-nitrosylases have been identified to date, such as Caspase 3 and glyceraldehyde-3-phosphate dehydrogenase (GAPDH). SNO-GAPDH has been reported to function as a nuclear transnitrosylase for many nuclear proteins such as sirtuin-1 (SIRT1) and histone deacetylase-2 (HDAC2), thus regulating nuclear gene transcription and cellular metabolism [35]. SNO-GAPDH has also been found to mediate the transnitrosylation of mitochondrial proteins in the heart [36]. SNO-caspase 3 can transnitrosylate the X-linked inhibitor of apoptosis (XIAP) to upregulate the apoptosis of neuronal cells [37].

Denitrosylases: Reciprocally, denitrosylation can be either an enzymatic or nonenzymatic reaction. Nonenzymatic denitrosylation can spontaneously occur under the nucleophilic attack of cytosolic reductants (such as ascorbate), which cleaves the S–N bond of the S-nitrosyl group [38]. Enzymatic denitrosylation is mainly catalyzed by three types of denitrosylases: thioredoxins (Trxs), GSNOR, and glutaredoxins (Grxs).

Trxs are a group of ubiquitous small redox proteins with a general disulfide reductase activity. There are two distinct classes of mammalian thioredoxins, Trx1 and Trx2. Trx1 is mainly localized in the cytoplasm, while Trx2 is restricted to mitochondria. Notably, Trx1 can also translocate to the nuclear compartment or be exported out of cells under certain conditions [39]. Trxs have a highly conserved CxxC motif in which two cysteines are separated by two other residues. The CxxC motif is located within a characteristic tertiary structure known as the thioredoxin fold, which consists of a four-stranded β-sheet and three flanking α-helices (β-α-β-α-β-β-α). The Trx fold core motif β-α-β-(CxxC)-α-β-β-α is essential for the redox catalysis of both intermolecular and intramolecular disulfide bonds, and it has been described as a rheostat in the active site since changes in the XX residues can dramatically alter the reduction potential and functional properties of the catalyst [40]. Trx, together with nicotinamide adenine dinucleotide phosphate hydrogen (NADPH) and thioredoxin reductase (TrxR) form the thioredoxin system. The thioredoxin system is a key antioxidant system in defense against oxidative stress through its disulfide reductase activity. TrxR is the only enzyme known to catalyze the NADPH-dependent reduction of oxidized Trxs to the reduced state, therefore serving as an essential component of the Trx system. The basic mechanism of the redox cascade of the Trx system is based on the flow of electrons from NADPH to TrxR to Trx to target proteins. Two forms of mammalian TrxRs have been identified so far: the cytosolic/nuclear and mitochondrial TrxRs (TrxR1 and TrxR2), which reduce Trx1 and Trx2, respectively. Trx1 catalyzes either the transnitrosylation or denitrosylation of target proteins. S-nitrosylation on the Cys73 of Trx1 mediates its transnitrosylating activity. The active disulfide bond between Cys32 and Cys35, which is reduced to free thiols by TrxR, catalyzes its denitrosylating activity [41]. It is worth noting that S-nitrosylated Cys73 can block the reduction of the Cys32-Cys35 disulfide bond and produce uncoupled Trx, thus preventing its denitrosylating activity [24]. Other than Trxs, several proteins have been identified to have a Trx fold and a Trx-like active-site sequence, such as protein disulfide isomerase (PDI) in the endoplasmic reticulum (ER). PDI consists of four domains with a thioredoxin fold that confers denitrosylase/transnitrosylase activities [42]. Another well-conserved member of the thioredoxin superfamily, the thioredoxin-related protein of 14 kDa (TRP14), has also been found to act as a denitrosylase [43].

GSNO denitrosylation is catalyzed by the main cellular denitrosylase S-nitrosoglutathione reductase (GSNOR), which is usually coupled with NADPH consumption. GSNO can be irreversibly reduced to glutathione disulfide (GSSG) by GSNOR in the presence of nicotinamide adenine dinucleotide (NADH) and GSH. GSSG can be further catalyzed by NADPH-dependent glutathione reductase (GR) to regenerate GSH [44,45]. Therefore, GSNOR controls the cellular levels of GSNO through the NADH-dependent denitrosylation of GSNO, leading to the indirect regulation of SNO-proteins and thus maintaining the thermodynamic equilibrium between SNO-proteins and SNO-LMW.

Additionally, under oxidizing conditions, the resulting GSSG from GSNOR denitrosylation can go through protein S-glutathionylation by reversibly forming a mixed disulfide (PSSG) between protein thiols (P-SH) and GSH [46]. The deglutathionylation of PSSG is catalyzed by the major deglutathionylating agents Grxs to give back reduced GSH molecules [47]. Aside from controlling the extent of protein S-glutathionylation, the S-denitrosylase activity of the Grx system has been highlighted recently. Grxs belong to the Trx superfamily, sharing both the Trx fold and thiol-disulfide oxidoreductase activity. The Grx system requires Grxs, GSH, GR, and NADPH. There are four isoforms of mammalian Grxs, namely Grx1, Grx2, Grx3, and Grx5, among which Grx1 and Grx2 are dithiol Grxs while Grx3 and Grx5 are monothiol Grxs. It was proposed by Ren et al. that Grxs can denitrosylase both low- and high-molecular-weight SNOs by either a dithiol mechanism that requires both active-site Cysteines or a monothiol mechanism that requires only the N-terminal cysteine at the active site [48]. Moreover, they found that GSH and Grx1 could synergistically denitrosylase SNOs, especially those GSH-stable SNOs, whereas monothiol Grxs only displayed denitrosylase activity when coupled with GSH. They also identified two substrates of Grx1, Caspase 3 and cathepsin B, that are involved in an intrinsic apoptotic pathway, suggesting an important role of Grx in apoptosis.

In summary, as shown in Figure 1, the cellular homeostasis of SNO is regulated by two main mechanisms—transnitrosylation and denitrosylation—through nonenzymatic and enzymatic reactions. The enzymatic reactions are mediated by S-nitrosylases and S-denitrosylases, which cooperatively control the steady-state cellular levels of SNO.

## 3. Applied Methods for the Detection and Quantification of SNO

The detection of SNO in biological samples tends to be difficult due to its particularly liable nature and low abundance. However, the development of a great number of reliable laboratory techniques has made it possible for the characterization, identification, and quantification of both SNO-proteins and SNO-sites. The applied methods are summarized below and also in Table 1.

### 3.1. X-ray Crystallography

Biophysical techniques such as X-ray crystallography and NMR spectroscopy provide powerful methods for the structural characterization of single, isolated SNO-proteins by high-resolution crystal structure analysis [49]. However, so far, only 14 X-ray structures of SNO-proteins, such as SNO-Hb, have been identified and curated in the Protein Data Bank (PDB) due to the great difficulty of obtaining SNO-proteins in sufficient quantities, as well as the unstable nature of the SNO bond upon radiation [50].

### 3.2. SNO-Specific Antibodies

The development of SNO-specific antibodies, such as SNO-cysteine- bovine serum albumin (BSA), provides a direct method to detect SNO-proteins from the immunohistochemical approach [51]. However, the specimen processing of the assay may compromise test specificity due to the lability of the SNO-bond. The specificity of currently available anti-S-nitrosocysteine (anti-CysNO) antibodies has also been brought into question [52].

### 3.3. Mass Spectrometry (MS)

MS-based technologies have been used to directly detect SNO-proteins, especially after protease digestion and purification, such as electrospray ionization MS (ESI-MS), a ‘soft ionization’ technique with a very gentle ionization process resulting in little or no cleavage of SNO-bonds [53]. The protonated SNO peptide ions shift +29 atomic mass units (amu) for each bound NO relative to unmodified ions.

### 3.4. NO-Based Assays

Indirect detection methods for the absolute quantification of protein-bound SNOs include a series of nitrite (NO_2_^−^) and NO-based assays that measure NO or NO_2_^−^ released upon S-NO cleavage [54]. S-NO bonds can be cleaved either heterolytically by divalent mercury (Hg^2+^) or homolytically by UV light. They are then subjected to various NO detection methods such as a Griess-Saville assay, ozone-based chemiluminescence, and fluorescence assays using 4,5-diaminofluorescein (DAF-2) or its analogous derivatives [55].

### 3.5. Biotin Switch Technique (BST)

Though NO-based assays allow for absolute quantification, they are unable to distinguish the source of each protein-SNO. In contrast, the biotin switch technique (BST) is a sulfur-based assay that has been most commonly used for the highly specific identification of individual SNOs. BST was initially developed by Jaffrey et al. in 2001 [56]. The initial BST consists of three principal steps. The initial step is blocking free thiols with methyl methanethiosulfonate (MMTS) via S-methylthiolation. Next, high concentrations of ascorbate are used to selectively break S–NO bonds and convert SNOs back to free nascent thiols. Last, free nascent thiols immediately undergo biotinylation. Though BST has been proved to be the method of choice, it can result in the false-positive identification of SNOs given that ascorbic acid is not a selective reductant of SNOs, and MMTS can exhibit cross-reactivity with sulfenic acids [57]. A false-positive can also arise from metal-ion contamination, the incomplete blocking of free thiols at the initial step, and the presence of indirect sunlight during biotinylation. Indirect sunlight during biotinylation results in the production of ascorbate-dependent artifacts and thus interferes with BST interpretation [52,58]. Therefore, setting both internal and external negative controls can improve the validity and reliability of BST results. Notably, photolysis by UV light before BST can eliminate SNO-dependent signals via the homolytic cleavage of SNO bonds, which has been proposed to be an ideal internal control that independently verifies assay specificity [59]. Various modifications of BST have been developed to improve test selectivity and specificity. For example, the classic biotin tag can be replaced by diverse tags such as fluorescent dyes (fluorescent switch technique), irreversible His-tags (His-tag switch technique), or thiol-reactive resins in SNO-resin-assisted capture (SNO-RAC) [60]. Parallel dual-labeling BST using both cysteine-reactive tandem mass tag (cysTMT) and iodoacetyl tandem mass tag (iodoTMT) in combination with MS have been reported to maximize S-nitrosylation site identification (SNO-SID) in mouse hearts due to greatly reduced labeling bias from the single tag-capture approach [61].

### 3.6. Derivatization-Based Combinations for SNO Identification

Recently emerging direct methods use alternatives to ascorbate-based enrichment strategies, such as gold nanoparticles (AuNPs) capture and organomercury resin capture (MRC). Such methods can achieve the identification of SNO-sites via the formation of stable SNO-conjugates that can be subjected to MS analysis [62]. Moreover, organophosphine-mediated bioorthogonal reactions of SNO have been explored as bioconjugation strategies. Triaryl-substituted phosphines can rapidly react with RSNO to generate azaylide intermediates. Azaylide intermediates can be manipulated into relatively stable conjugates via fast intramolecular reactions such as the reductive ligation reaction, the bis-ligation reaction, the reductive elimination reaction, and one-step disulfide formation. The one-step disulfide formation has been applied to SNO direct labeling, in which biotin-linked phosphine that incubated with SNO-proteins form a stable disulfide linkage with biotin and subsequently generate biotin-conjugates [63]. The bis-ligation reaction of endogenous LMW-RSNOs, particularly that of GSNO with triarylphosphine-thiophenyl ester, generates disulfide-iminophosphorane that can be detected by LC-MS for the quantitative analysis of GSNO in the physiological system [64].

The bioorthogonal cleavable-linker and switch technique (Cys-BOOST) combined with LC-MS/MS has recently been reported for the quantitative analysis of SNO sites and the detection of consensus motifs [65]. In Cys-BOOST, all free thiols are removed by iodoacetamide (IAA), and SNO peptides are switched with IAA-alkyne after being reduced by ascorbate [66]. IAA-alkyne labeled peptides are subjected to selective conjugation with a Dde-biotin-azide cleavable linker following proteolysis and amine-reactive TMT labeling. Dde-biotin azide, after cleavage, releases the biotin tag, which binds to streptavidin. Cys-BOOST has greatly improved the overall coverage of the proteome mapping of SNO-proteins with significantly higher sensitivity and specificity. However, one general flaw of bio-orthogonal labeling with phosphines is that the aqueous solubility of phosphine is very low and thus hinders the detection of biological SNOs in aqueous systems. This flaw has motivated the development of water-soluble Tris (2,4-dimethyl-5-sulfophenyl) phosphine trisodium salts (TXPTS) as biomarkers for protein S-nitrosylation [67].

### 3.7. Isotope Coded Affinity Tag (ICAT)

The application of iodoacetamide analogs such as isotope-coded affinity tag (ICAT) and second-generation cleavable ICAT (cICAT) have been intensively used for both the qualitative and quantitative proteomic analysis of SNO-peptides with LC-MS/MS [68]. The structure of the ICAT consists of a thiol-reactive biotinylated iodoacetamide (BIAM) and a linker containing either light (^12^C) or heavy (^13^C) isotopes. The isotopes can differentially label nascent free thiols to determine the relative abundance of proteins in paired samples. A cICAT contains an additional acid-cleavable linker that can remove the biotin tag before MS analysis to improve MS spectra.

### 3.8. Cysteine-Reactive Tandem Mass Tag (cysTMT)

Other than identifying specific SNO-proteins and SNO-SID, cysteine-reactive tandem mass tag (CysTMT) labeling combined with large-scale MS analysis has been used to determine the SNO-occupancy of each modifiable Cys in the myocardium for measurement of thiol-reactivity to SNO-modification [69]. SNO occupancy is defined as the ratio or percentage of a given protein modified by SNO, which has rarely been studied due to the technical difficulties in measurement.

## 4. SNO in Cardiac Pathogenesis and Protection

The heart is one of the most affected organs by SNOs, and a large number of SNOs have been reported in cardiomyocytes [70,71]. SNOs in the heart exert many functions, including vasodilation, anti-inflammatory effects, anti-thrombotic effects, oxygen homeostasis, the regulation of angiogenesis, apoptotic/necrotic cell death, and intracellular Ca^2+^ homeostasis. Aberrant or dysregulated SNO-dependent NO signaling has been linked to heart diseases and related conditions such as myocardial ischemia, heart failure, and atrial fibrillation [72]. However, the molecular mechanisms underlying these discoveries are still far from being fully elucidated. We synopsize the up-to-date studies of S-nitrosylation in various cardiac diseases in humans and animal models, and we discuss its therapeutic potential in cardiac protection in the following sections; relative references are summarized in Table 2.

### 4.1. Protein S-Nitrosylation in the Heart

The studies related to SNO protein in the hearts are concentrated in the ischemic myocardial injury. Several SNO proteins have been found to play a protective role against ischemia-reperfusion (IR) injury. On the contrary, some increased SNO proteins have been found to be associated with cardiac hypertrophy.

#### 4.1.1. Cardiac Protection of SNO in Ischemic Myocardial Injury

Increased SNO formation has been linked to cardioprotection against ischemic myocardial injury. Nadtochiy et al. investigated the protective efficacy of the mitochondrial S-nitrosating agent S-nitroso-2-mercaptopropionyl glycine (SNO-MPG) in vivo using murine models of myocardial ischemia (MI) with permanent left anterior descending artery (LAD) ligation [73]. Their results showed that in vivo SNO-MPG exhibited a similar extent of cardioprotective effects against IR injury as ischemia precondition (IPC), a golden standard strategy of cardiac protection. They detected that S-nitrosylated proteins in IPC mitochondria widely overlapped with those in SNO-MPG-treated mitochondria. Moreover, the cardioprotection conferred by either SNO-MPG or IPC requires functionally intact mitochondrial complex I. Chouchani et al. used a mitochondrial selective S-nitrosating agent, MitoSNO, to explore the potential mechanism underlying the cardioprotective effects of mitochondrial S-nitrosylation in vivo during the reperfusion phase of acute murine MI [74]. They identified the S-nitrosylation of Cys39 on the NADH dehydrogenase 3 (ND3) subunit of mitochondrial complex I to be responsible for MitoSNO’s cardioprotective effects on ischemic myocardium during reperfusion. They suggested that the S-nitrosylation of ND3 Cys39 could mediate the reversible inhibition of complex I activity by disrupting its interaction with ubiquinone and decreasing ROS production. Their discovery of the unique local environment of the ND3 subunit within complex I provided the structural basis for exposing the occluded Cys39 to SNO modification during ischemia. SNO-Cys39 then switches complex I activity to a low state at reperfusion. Based on these findings, Methner et al. further explored the long-term cardioprotective effects of Mito-SNO against post-infarct heart failure using an in vivo mouse model of MI achieved by LAD ligation [75]. Their results showed that persistent infusion of Mito-SNO infusion greatly reduced the infarct size and troponin level in MitoSNO-treated hearts at 24 h post-reperfusion. Moreover, cardiac functions in MitoSNO-treated hearts at 28 days post-MI were also significantly improved, indicating that acute Mito-SNO administration could have both short-term and long-term protective effects for the heart. Additionally, the administration of Mito-SNO did not influence hemodynamic parameters, which made it potential for the clinical application of Mito-SNO. Kohr et al. investigated the role of the S-nitrosylation of tripartite motif-containing protein 72 (TRIM72) at cysteine 144 (C144) in Langendorff-perfused mouse hearts [76]. TRIM72, also known as Mitsugumin-53 (MG53), is a membrane repair protein located predominantly at cardiac and skeletal muscle that can be activated upon oxidative stress or ischemia/reperfusion (I/R) injury-induced membrane damage and, subsequently, translocate to the injured site to facilitate membrane repair as a scaffold [77]. Their results showed that SNO-TRIM72 was significantly increased with IPC in Langendorff-perfused mouse hearts and that SNO could shield C144 of TRIM72 from irreversible oxidative damage, thus protecting against I/R-induced TRIM72 degradation and cell death in Langendorff-perfused mouse hearts. These results provided insights into the use of recombinant MG53 (rhMG53) as a biomarker and a potential therapeutic for myocardial ischemic injury [78].

Studies have also been conducted to explore the mechanism of SNO involved in cardiac protection. It has been demonstrated that aside from entering the cell nucleus as a nuclear transnitrosylase in the form of SNO-GADPH, GADPH can also be imported into the mitochondria and modulate mitophagy and apoptosis after myocardial IR injury [79]. Kohr et al. investigated the interaction between SNO-GADPH and mitochondrial proteins in mice heart with/without preceding myocardial IPC-IR injury [36]. Their results showed a significant increase of GADPH in the mitochondrial fraction of IPC hearts without increasing the total GADPH level. Purified GAPDH and SNO-GAPDH after trypsin digestion were detected in the mitochondrial matrix, thus indicating the ability of GAPDH to enter the mitochondrial matrix. Importantly, SNO-GAPDH could rapidly increase mitochondrial protein S-nitrosylation, which was found to be in correlation with mitochondrial SNO levels. The findings indicate that SNO-GADPH acting as a mitochondrial transnitrosylase mediating the transnitrosylation of heat shock protein 60 (Hsp60) and acetyl-CoA Acetyltransferase 1 (ACAT1).

Sun et al. reported that methyl-β-cyclodextrin (MβCD) treatment could disrupt myocardial caveolae structure at the sarcolemma [80]. The disruption of myocardial caveolae resulted in abrogated the IPC-induced cardioprotection and IPC-Induced increase of protein S-nitrosylation in Langendorff-perfused mouse hearts. They concluded that MβCD exerted its inhibitory effects by dissociating the colocalization of caveolin-3 with eNOS in myocardial caveolae, thus dramatically interfering with the critical compartmentalization in regulating the eNOS signaling pathways. Based on these findings, Sun et al. further explored the association of caveolar structures in the context of the cardioprotective signaling in Langendorff-perfused mouse hearts with/without IPC [81]. Two distinct mitochondrial subpopulations have been identified in the myocardium, namely subsarcolemmal mitochondria (SSM), situated directly beneath the sarcolemmal membrane, and interfibrillar mitochondria (IFM), which are distributed between myofibrils. Their results showed that SSM exhibited a higher SNO level than IFM at baseline (control), and an increase of SNO content was only observed in SSM other than IFM in IPC hearts. A co-immunoprecipitation (co-IP) analysis revealed that only eNOS and caveolin-3 were associated with SSM, and both protein levels were elevated in IPC hearts. The SSM of caveolin-3^−/−^ mouse hearts subjected to IPC treatment exhibited a loss of cardioprotection and the rise of SNO induced by IPC. Thus, SSM could be considered a favored target of the caveolae/eNOS/NO/SNO signaling in IPC-induced cardioprotection. Notably, the gap junction (GJ) protein connexin 43 (Cx43) expressed in myocardial mitochondria exists exclusively at the inner membrane of SSM and serves as a marker for SSM [82]. Mitochondrial Cx43 (mtCx43) has been considered an essential component of cardiac preconditioning. It has long been linked to IPC-induced cardioprotection and attenuated reperfusion-induced reactive oxygen species (ROS) production, as well as the opening of mitochondrial K_ATP_ channels (mK_ATP_) [83]. Soetkamp et al. investigated the regulatory effects of nitrite-induced SNO of mtCx43 during I/R injury via pharmacological preconditioning with sodium nitrite (NaNO_2_) in the Langendorff-perfused mouse heart model [84]. Their results showed that NO-mediated increase in ROS production was driven by a proton gradient across the inner membrane. These increases in SSM could all be blocked by a Cx43 hemichannel blocker, carbenoxolone (CBX). These results together suggest that the SNO of mtCx43 plays an important role in mediating mitochondrial permeability, K^+^ influxes, and ROS formation, thus participating in the cardioprotective signal transduction cascade.

Sun et al. also set foot in clarifying the participation of the two major NO-dependent signaling pathways, soluble guanylyl cyclase/cyclic guanosine monophosphate/protein kinase G (sGC/cGMP/PKG) and protein S-nitrosylation, in NO-mediated acute IPC-induced cardioprotection in Langendorff-perfused mouse hearts [85]. Their results indicated that IPC-induced cardioprotection depends on SNO signaling, not sGC/cGMP/PKG signaling. Aside from IPC, “conditioning” the heart to tolerate acute I/R injury can also be applied after the onset of sustained MI, which is called ischemic postconditioning (PostC). PostC is usually performed at the beginning of the reperfusion by intermittently interrupting early reperfusion with brief coronary occlusions, which has been reported to be more cardioprotective against myocardial I/R injury than IPC [86]. Tong et al. investigated cardiac protein S-nitrosylation after PostC using the Langendorff-perfused mouse heart model [87]. Their results showed that PostC hearts exhibited significantly higher myocardial protein SNO levels than I/R-controls, which could be abolished by L-N^G^-Nitroarginine methyl ester (L-NAME), a constitutive NO synthase inhibitor. Furthermore, PostC-induced cardioprotection could be comparably mimicked by an NO donor S-nitroso-N-acetyl-D, L-penicillamine (SNAP). In the meantime, PostC-induced cardioprotection could not be reversed by inhibitors of the sGC/cGMP-dependent NO signaling pathway. These results suggest the involvement of NO-mediated SNO signaling instead of sGC/cGMP/PKG signaling in PostC-induced cardioprotection. Sun et al. explored the cross-talk between S-sulfhydration (SSH) mediated by hydrogen sulfide (H_2_S) and S-nitrosylation in PostC hearts using the Langendorff-perfused mouse hearts and I/R protocol [88]. Their results showed that additive cardioprotection was found to be simultaneously induced in PostC hearts by using sodium hydrosulfide (NaHS) and NO donor SNAP together, and NaHS-induced cardioprotection as well as its synergistic increase with SNAP was considerably attributed to an increase in SNO.

#### 4.1.2. SNO Involved in Pathogenesis of Myocardial Hypertrophy

Tang et al. explored the role of the S-nitrosylation of muscle LIM protein (SNO-MLP) involved in the pathogenesis of myocardial hypertrophy. They examined the SNO-MLP level in myocardial samples acquired from patients diagnosed with myocardial hypertrophy, transverse aortic constriction (TAC) mice, angiotensin II/phenylephrine-treated neonatal rat cardiomyocytes, and TLR3 (Toll-like receptor 3) knockout mice. A significant increase of SNO-MLP on Cysteine 79 (Cys79) was observed in hypertrophic myocardium. Increased SNO-MLP promoted the binding of Cys79 with downstream TLR3, followed by TLR3-mediated RIPK3 (receptor-interacting protein kinase 3) and NLRP3 (NOD-like receptor pyrin domain containing 3) inflammasome activation [89]. Therefore, the denitrosylation of MLP could serve as a potential therapeutic target for myocardial hypertrophy and heart failure.

Vielma et al. explored the relationship between NOS isoforms and isoproterenol (ISO)-induced S-nitrosylation by using selective pharmacological inhibitors of NOS-1 and NOS-3 [90]. Their results showed that NOS1 blockade significantly reduced basal cardiac contractility and inotropic response to βA stimulation. The NOS1 blockade also greatly reduced the increment in basal S-nitrosylation of key Ca^2+^ handling protein RyR2, suggesting a direct association between NOS1-dependent protein S-nitrosylation and contractility. Moreover, ROS inhibition with Tempol, a free radical scavenger, reduced not only protein S-nitrosylation and cardiac contractility but also inotropic responses to ISO. Tempol strongly mimicked the effects of NOS-1 inhibitors, indicating that ROS production could link between NOS1-mediated S-nitrosylation and βA stimulation.

### 4.2. GSNO in the Heart

Like the SNO proteins, SNO-LMWs have also been heavily studied in the stressed heart, particularly the GSNO. Since GSNOR plays a crucial role in GSNO denitrosylation and in maintaining the equilibrium between SNO-proteins and GSNO, various studies have been conducted in the hearts with GSNOR knockout (KO) and transgenic (TG) mouse models.

#### 4.2.1. GSNO/GSNOR in Ischemic Hearts

Sun et al. showed that the effects of IPC on protein S-nitrosylation and cardioprotection against myocardial IR injury were mimicked by GSNO [91]. They reported that GSNO not only greatly decreased calcium transients during IR but also altered the activities of SERCA2a and proteins involved in mitochondrial energetics, such as alpha-ketoglutarate dehydrogenase (α-KGDH) and F1-ATPase. The pathological opening of the mitochondrial permeability transition pore (mPTP) has been intensely linked to myocardial I/R injury and apoptotic cell death. Mitochondrial cyclophilin D (CypD) has been found to be a key regulator of the mPTP and to contain a redox-sensitive residue Cys-203 that can be S-nitrosylated in the heart after treatment with GSNO [71]. Similar results were also observed by Nguyen et al. in Langendorff-perfused hearts [92]. Their results showed that GSNO treatment significantly reduced H_2_O_2_-induced mPTP opening in WT mouse embryonic fibroblasts (MEFs). The demonstrated inhibition of mPTP opening with GSNO treatment confirmed the cytoprotective role of S-nitrosylated CypD. Additionally, CypD^−/−^ MEFs transfected with either WT CypD or C203S-CypD generated by the site-directed mutation of Cys 203 of CypD to a serine showed a similar level of inhibition on mPTP opening, suggesting that the inhibitory effects on the mPTP opening by SNO of Cys-203 was similar to that of C203S-CypD or CypD deletion. Amanakis et al. recently identified Cys-202 of CypD as a redox-sensitive site for multiple PTMs, including S-nitrosylation [93].

Since GSNOR is key to maintain the equilibrium between SNO-proteins and GSNO, it has been extensively studied and is considered an essential element in cardiovascular health and diseases. Lima et al. found that normoxic myocardium of GSNOR KO mice after an LCA (left coronary artery) ligation exhibited a significantly improved cardiac performance compared to a control group [94]. Their results showed that the constitutive S-nitrosylation of hypoxia-inducible factor 1-alpha (HIF-1α) by endogenous GSNO could lead to increased HIF-1α transcription. S-nitrosylated HIF-1α binding to downstream vascular endothelial growth factor (VEGF) resulted in promoted myocardial angiogenesis against ischemic attack. Beigi et al. reported that subcellular colocalization of GSNOR with NOS1 and RyR2 inside cardiac sarcoplasmic reticulum (SR) could provide a structural basis for the S-nitrosylation/denitrosylation of RyR2 [95]. They also emphasized that the aberrant denitrosylation of RyR2 could lead to increased diastolic Ca^2+^ leak in GSNOR^−/−^ cardiomyocytes and, subsequently, compromised cardiac contractility. Hatzistergos et al. found that, aside from a generally improved cardiac performance, post-MI adult heart from GSNOR^−/−^ knockout mice also exhibited increased regenerative and proliferative activity of adult cardiac progenitors, mesenchymal stem cells, and cardiomyocytes compared with WT controls [96]. This discovery revealed the cell-cycle-dependent effects of GSNOR and paved the way for the potential development of therapeutic targets against endogenous denitrosylases.

#### 4.2.2. GSNO/GSNOR in Cardiac Hypertrophy

Irie et al. explored the role of β-adrenergic receptor (β-AR) signaling in the pathophysiology of cardiomyocyte hypertrophy and heart failure in mice [97]. Their results showed that enhanced denitrosylation in GSNOR-Tg (GSNOR overexpression) mice could inhibit the increase of intracellular Ca^2+^, suppressing the Ca^2+^-calcineurin signaling and leading to the induction of left ventricular hypertrophy (LV) and dysfunction under chronic ISO challenge. Additionally, GSNOR overexpression or pharmacologic NOS inhibition was found to block the increase of the S-nitrosylation of phospholamban (PLN), NCX (Na^+^/Ca^2+^ exchanger), and cTnC (cardiac troponin C) in the heart upon ISO stimulation. In particular, their results showed that the formation of a functionally inactive PLN pentamer under ISO stimulation required not only the phosphorylation of inhibitory monomeric PLN at Ser^16^ and Thr^17^ but also the S-nitrosylation of PLN at Cys^36^ or Cys^41^, while the S-nitrosylation of cTnC at Cys^84^ led to decreased βAR-induced responsiveness.

Mori et al. reported that the specific overexpression of GSNOR in cardiomyocytes (GSNOR-CMTg) exhibited mitigated LV dysfunction and hypertrophy compared to WT controls during chronic pressure overload [98]. Pressure overload was induced by TAC in wild-type, GSNOR^−/−^, and GSNOR-CMTg mice. GSNOR^−/−^ mice, on the other hand, showed markedly worsened LV hypertrophy than WT controls, indicating that increased denitrosylation by GSNO, although not able to prevent LV dysfunction, could reduce the degree of cardiac hypertrophy during chronic pressure overload. GSNOR-Tg mice were also reported to be cardioprotective against sepsis-induced myocardial depression under a lipopolysaccharide (LPS) challenge [99].

### 4.3. Sex Differences in SNO-Mediated Cardiac Effects

It is generally believed that women tend to develop heart diseases ten years later than men and have significantly better overall cardiac function and survival against heart diseases. The selective activation of estrogen receptor-β (ER-β) has been closely linked to cardioprotection observed in females following MI [100]. Lin et al. explored ER-β-mediated cardioprotection during I/R injury in an ovariectomized (OVX) mice model [101]. The OVX mice were treated with an ER-β selective agonist, 2,2-bis (4-hydroxyphenyl)-propionitrile (DPN), 17β-estradiol (E2), or vehicle. Their results showed that DPN and E2 treatment could induce cardioprotective effects to similar extents during I/R injury and increase overall protein S-nitrosylation levels in the ovariectomized mouse model. Moreover, the DPN-induced cardioprotection could be abolished by the pretreatment of an NOS inhibitor, L-NAME, in I/R hearts. However, DPN-induced cardioprotection was not observed in the ovariectomized estrogen receptor (ER) β KO mouse hearts. These results suggested that DPN-induced cardioprotection relied on both the activation of ER-β and NO/SNO signaling.

Shao et al. investigated the cardioprotective effects of pharmacologic preconditioning in male and female hearts using an adenosine A1 receptor agonist N6-cyclohexyl adenosine (CHA) in a murine model of I/R injury [102]. They identified a number of consistently S-nitrosylated proteins after comparing CHA-induced SNO-proteins in male and female hearts, which strongly suggest the cardioprotective role of SNOs. SNO on the dihydrolipoyl dehydrogenase of the alpha-KGDH complex was considered to be responsible for the significantly decreased ROS production in female hearts, serving as a potential mechanism to explain the gender-related differences in the face of heart diseases.

To clarify the molecular signature of failing hearts, Menazza et al. employed oxidation RAC (Ox-RAC) and SNO-RAC in the detection of protein oxidation and S-nitrosylation levels in left ventricular samples acquired from biopsies in dilated cardiomyopathy (DCM) and heart failure (HF) as well as from healthy male and female donors [103]. The sex-specific differences of SNO levels in DCM hearts detected within failing male hearts showed a greater SNO increase than failing female hearts. Nonfailing hearts, on the other hand, exhibited higher levels of SNO in females than in males, and the majority of them were from mitochondria.

Based on their previous findings that female hearts exhibit higher protein-SNO levels and significantly increased GSNOR activity at baseline compared to male hearts, Casin et al. investigated the sex-dependent cardioprotection of GSNOR against I/R injury in males and female mice with both ex vivo and in vivo models of myocardial I/R injury [104]. Their results indicated that both acute GSNOR inhibition by a potent GSNO-R inhibitor N6022 and the genetic deletion of GSNOR (GSNOR^−/−^) in male hearts exhibited significantly reduced infarct size and enhanced post-ischemic functional recovery. In contrast, female hearts yielded the opposite results, with a worsened infarct size and functional recovery post-MI. Moreover, the cardioprotective effects induced by a GSNOR blockade in male hearts can be attributed to the increased pre-ischemic SNO levels of ND3 Cys39 and a substantially decreased post-ischemic H_2_O_2_ production. Additionally, they found that GSNOR inhibition could impair GSNOR metabolism as a formaldehyde dehydrogenase in the heart, especially in female hearts where post-ischemic free formaldehyde levels were significantly elevated after N6022 treatment. Excessive mitochondrial formaldehyde was found to exert a detrimental impact on female hearts, which could be rescued by an aldehyde dehydrogenase (ALDH) agonist Alda-1.

### 4.4. Therapeutic Potential of SNO

Inorganic nitrites and nitrates, such as sodium nitrite that was used as pharmacological preconditioning in the above experiment, have long been reported to have cardioprotective effects against myocardial I/R injury [105]. Additionally, the clinical use of organic nitrates such as sodium nitroprusside has been shown to exhibit antiarrhythmic effects associated with alterations in cardiac electrophysiology [106]. Kovacs et al. used preclinical large animal models to examine whether the acute administration of inorganic sodium nitrite could protect against the I/R-induced severe ventricular arrhythmias and whether SNO took part in this protective mechanism [107]. Anesthetized dogs were infused intravenously with either saline or sodium nitrite (NaNO_2_) before and during a 25 min period of LAD occlusion (NaNO_2_-PO group) and 10 min prior to reperfusion (NaNO_2_-PR group). Their results showed that both NaNO_2_-treated groups exhibited a significantly attenuated severity of ischemia following coronary occlusion compared to sham controls, with the NaNO_2_-PR group gaining the highest survival rate of 92%. Meanwhile, sodium nitrite also markedly reduced the total number of ventricular premature beats (VPBs) and the incidence and number of ventricular tachycardia (VT) episodes. SNO proteins and peptides were remarkably increased in the NaNO_2_-PR group but not in the control and NaNO_2_-PO groups, although the results were not statistically significant. Their research provided the first evidence that the inorganic sodium nitrite could protect against the I/R-induced severe ventricular arrhythmia. This cardioprotective effect was associated with increased protein SNO levels only when sodium nitrite was administered prior to reperfusion. More preclinical research will be expected based on the increasing studies in this field.

### 4.5. SNO in other Cardiac-Associated Diseases

A large number of current discoveries regarding S-nitrosylation have emphasized the heart itself. Meanwhile, the role of S-nitrosylation in other diseases that may cause cardiomyopathies, such as hypertension, atherosclerosis, stroke, and diabetes, have been also studied. However, the underlying molecular mechanisms remain largely undetermined.

Choi et al. explored the role of S-nitrosylation in hypertension by using the murine angiotensin II (AngII) infusion model [108]. They reported that an increased S-nitrosylation induced by either TrxR inhibition or CysNO could decrease the acetylcholine (ACh)-mediated vascular relaxation of aortic rings in Ang II/phenylephrine-treated mice. Their study was the first to suggest the relevance of increased S-nitrosylation to impaired aortic relaxation in AngII-induced hypertensive mice. Jeong et al. reported that the nNOS-mediated S-nitrosylation of transglutaminase 2 (TG2) in LV myocytes could impede fatty acid utilization and fatty acid-induced cardiac contractility in Ang II-induced hypertensive mice [109]. Neto-Neves et al. recently reported that the oral administration of NaNO_2_ into two-kidney, one-clip (2K1C) renovascular hypertensive rats could restore cardiac performance along with increased protein S-nitrosylation in LV myocardium [110].

In addition, Li et al. found that hyperhomocysteinemia (HHcy) could aggravate atherosclerosis by upregulating GSNOR activity in T cells and switching the S-nitrosylation of protein kinase B/Akt Cys 224 to the phosphorylation of Akt Ser473 [111]. The phosphorylation of Akt Ser473 led to the GSNOR-dependent activation of Akt in the phosphoinositide-3-kinase-protein kinase B/Akt (PI3K-PKB/Akt) pathway. Akt activation further increased inflammatory cytokine secretion and T-cell proliferation. Therefore, GSNOR^−/−^ apolipoprotein E (ApoE)^−/−^ double knockout mice subjected to HHcy challenge exhibited much less T-cell activation and a significantly mitigated amount of total lesions of HHcy-induced atherosclerosis compared with GSNOR^+/+^ApoE^−/−^ mice. Lin et al. reported that the administration of an H_2_S donor, NaHS, could generate a significant anti-atherosclerotic effect against atherosclerotic aortic lesions and the accumulation of lipid-laden macrophages in apoE^−/−^ mice subjected to an atherogenic diet [112]. Furthermore, the blockade of cystathionine γ-lyase (CSE), an endogenous H_2_S-producing enzyme, resulted in the opposite effects to NaHS. The anti-atherosclerotic effect was attributed to increased plasma NO and protein S-nitrosylation in aorta vascular smooth muscle cells (VSMCs). Wang et al. further examined the role of H_2_S in ApoE^−/−^ CSE^−/−^ knockout mice [113]. They confirmed the protective effects of endogenous H_2_S against atherogenesis through upregulating protein S-nitrosylation. Their study provided insights into the therapeutic potential of H_2_S-releasing drugs that could bypass the defects of NaHS.

Furthermore, Shi et al. reported that, during acute ischemic stroke, the excessive activation of neuronal N-methyl-D-aspartate receptors (NMDARs) could result in NO overproduction and the subsequent S-nitrosylation of tyrosine phosphatase-2 (SHP-2) [114]. S-nitrosylated SHP-2 (SNO-SHP-2) could inhibit its phosphatase activity, which led to the downregulation of the extracellular signal-regulated kinase 1/2 (ERK1/2) pathway, thus contributing to increased excitotoxic neuronal damage. S-nitrosylation protein targets such as matrix metalloproteinase 9 (MMP-9) and the phosphatase and tensin homolog (PTEN) have been implicated in both acute stroke and chronic neurodegenerative diseases [115]. Okamoto et al. observed an increase of the S-nitrosylation of myocyte enhancer factor 2 (MEF2) in the middle cerebral artery (MCA) occlusion model of stroke in mice [116]. They suggested that the aberrant S-nitrosylation of MEF2 (SNO-MEF2) could compromise the antiapoptotic B-cell lymphoma-extra large (Bcl-xL) pathway to the extent that it could lead to cerebrocortical neuron death in stroke.

Besides these cardiac-associated vascular diseases, abnormal S-nitrosylation has also been related to vascular lesions and insulin resistance associated with type 2 diabetes and obesity. Wadham et al. reported that the exposure of endothelial cells to high glucose could significantly reduce the S-nitrosylation of proteins involved in regulating vascular functions, such as eNOS [117]. Moreover, the inhibitory effects of high glucose could be fully reversed by ROS inhibitors, suggesting a significant relationship between protein S-nitrosylation and hyperglycemia-induced vascular oxidative stress. Their study contributed to a better understanding of hyperglycemia-induced endothelial dysfunction in diabetes. Yasukawa et al. reported that NO donor-induced S-nitrosylation at Cys224 in the skeletal muscle of diabetic (db/db) mice could lead to the PI3K-independent inactivation of Akt/PKB in insulin signaling, contributing to insulin resistance in type 2 diabetes [118]. Qian et al. reported that obesity and diabetes could increase the S-nitrosylation of key lysosomal proteins and decrease GSNOR-mediated denitrosylation in the livers of mice on a high-fat diet and genetically obese (ob/ob) mice [119]. Diminished denitrosylation resulting from GSNOR deficiency led to increased lysosomal nitrosative stress and defective autophagy, impairing hepatic insulin sensitivity and contributing to hepatic insulin resistance. Ovadia et al. detected an overall increase of protein S-nitrosylation levels in adipose tissue acquired from nutritional high fat-fed (HFF) mice, ob/ob mice, and humans [120]. Moreover, they found that NO donor-induced S-nitrosylation could directly impede the anti-lipolytic action of insulin upon the activation of phosphodiesterase 3 (PDE3B). They suggested that PDE3B could serve as a specific target for the S-nitrosylation of adipose tissues in obesity. The body mass index (BMI) has been most commonly used to screen overweight or obesity. Though S-nitrosylation has been associated with overweight/obesity, there has not been a direct link between SNO regulation and BMI.

## 5. Conclusions and Future Direction

Increasing evidence has demonstrated that S-nitrosylation is an essential PTM in maintaining normal cardiac cellular function and plays a key role in regulating the pathogenesis of cardiac diseases or cardiac stresses. S-nitrosylation is becoming a potential target of cardiac protection that could provide new avenues for developing new therapeutic strategies.

The field of protein S-nitrosylation has proved to be rapidly evolving over the last several years. Despite the importance of this PTM in various cellular functions, SNO regulation in the heart and the mechanisms underlying the cardiac pathogenesis and protection remain largely unknown. Several key aspects need to be considered in the future directions. First, despite the considerable progress made in identifying SNO-proteins and motifs in vitro, the real-time detection of cellular SNO in the heart is still a big challenge. More reliable and in situ detective strategies need to be developed, especially in vivo measurements. The development of these advanced techniques would significantly contribute to our current understanding of this emerging paradigm redox signaling, especially in the setting of heart diseases. Second, extensive research efforts are needed to elucidate the complex molecular mechanisms underlying the bioactivity of S-nitrosylation, including the regulations on its formation, spatiotemporal specificity, homeostasis, and the association with cellular redox status and various stresses, as well as the cross-talk with other PTMs in proteins. Third, the identification of specific SNO proteins and their biological function in the heart will help develop new knowledge for the treatment and prevention of myocardial diseases and apply these findings to clinical and translational medicine. Further studies in these fields would bring new insights into preventing and treating heart diseases.

## Figures and Tables

**Figure 1 antioxidants-09-01051-f001:**
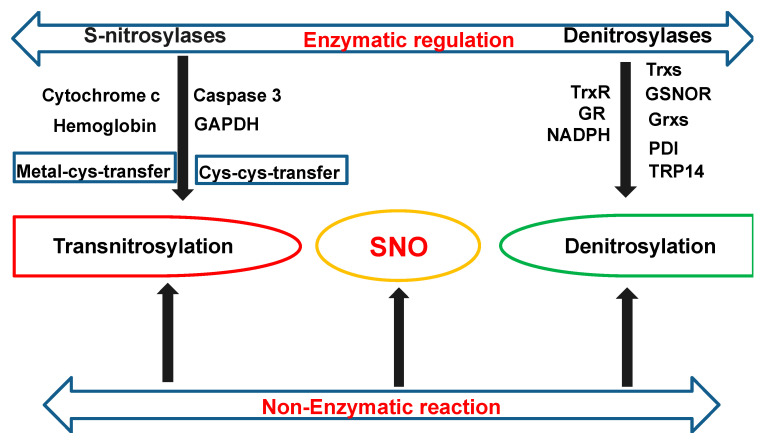
The scheme of main enzymatic regulations on S-nitrosothiol (SNO) homeostasis. The cellular homeostasis of SNO is regulated by two main mechanisms, transnitrosylation and denitrosylation, through nonenzymatic and enzymatic reactions. The enzymatic reactions are mediated by S-nitrosylases and S-denitrosylases, which cooperatively control steady-state cellular levels of SNOs. Enzymatic SNO formation can be achieved via a Metal-to-Cys transfer by metalloproteins, such as hemoglobin and cytochrome c, or via Cys-to-Cys transnitrosylase such as SNO-caspase 3 and SNO- glyceraldehyde 3-phosphate dehydrogenase (GADPH). Protein denitrosylation is mainly catalyzed by three major systems, the thioredoxins (Trxs) system, the GSNOR (S-nitrosoglutathione reductase) system, and the glutaredoxin systems (Grxs). Both Trxs and Grxs belong to the Trx superfamily. A few proteins other than Trxs and Grxs from the same family have also been identified as S-denitrosylases, such as PDI (protein disulfide-isomerase) and TRP14 (a thioredoxin related protein of 14 kDa).

**Table 1 antioxidants-09-01051-t001:** Summarization of methods for detection, identification, and quantitation of SNO.

Methods	Characters	References
X-ray crystallography and NMR spectroscopy	Structural-based determination Challenging to achieve due to the lability of SNO-bonds	[49,50]
SNO-specific antibodies	Immunohistochemical approach Commercially available antibodies are questionable	[51,52]
MS/LC-MS/ESI-MS	Approach based on mass-to-charge ratios ESI-MS: soft ionization technique LC-MS: analyze SNOs in the physiological system	[53]
AuNP capture MRC Cys-Boost	Derivatization-based assays Bioconjugation strategies and bioorthogonal reactions	[62,63,64,65,66,67]
Griess-Saville assay Ozone-based chemiluminescence Fluorescence assays (DAF-2 or its analogous derivatives)	NO-based assays NO or NO_2_^−^ released upon SNO cleavage	[54,55]
BST SNO-SID Fluorescent switch technique His-tag switch technique SNO-RAC Parallel dual-labeling BST CysTMT and IodoTMT	BST-based assays SNO-SID: BST in combination with MS CysTMT: determine SNO-occupancy	[56,57,58,59,60,61,69]
ICAT/cICAT	Quantitative proteomics	[68]

SNO: S-nitrosothiols; ESI-MS: electrospray ionization MS; AuNPs: gold nanoparticles; MRC: organomercury resin capture; Cys-BOOST: bioorthogonal cleavable-linker and switch technique; DAF-2: 4,5-diaminofluorescein; BST: biotin-switch technique; SNO-SID: SNO site identification; SNO-RAC: SNO-resin-assisted capture; cysTMT: cysteine-reactive tandem mass tag; IodoTMT: iodoacetyl tandem mass tag; ICAT: isotope-coded affinity tag; cICAT: cleavable ICAT.

**Table 2 antioxidants-09-01051-t002:** Summarization of studies on SNO involved in cardiac pathogenesis and protection.

Reference	Studied Model	Research Focus
[73]	Murine I/R-IPC	The cardioprotection of mitochondrial SNO-MPG against MI
[74]	Murine I/R	The acute cardioprotective effects of Mito-SNO against MI
[75]	Murine I/R	Long-term cardioprotective effects of Mito-SNO against MI
[76,77,78]	Murine I/R-IPC	The role of SNO-TRIM72 at cysteine144 against MI
[36,79]	Murine I/R-IPC	SNO-GADPH interacts with mitochondrial SNOs against MI
[80]	Murine I/R-IPC	The role of myocardial caveolae in eNOS/NO/SNO cardioprotective signaling against MI
[81,82]	Murine I/R-IPC model	SSM as a major target of eNOS/NO/SNO signaling
[83,84]	Murine I/R-IPC	The effects of nitrite-induced SNO of mtCx43 against MI
[85]	Murine I/R-IPC	SNO signaling vs. sGC/cGMP/PKG signaling in IPC-induced cardioprotection
[86,87]	Murine I/R-PostC	Cardiac SNO-proteins in PostC-induced cardioprotection
[88]	Murine I/R-PostC	S-sulfhydration and S-nitrosylation Crosstalk in PostC hearts
[89]	Murine TAC and lSO- HCM; TLR3 KO mice	SNO-MLP in the pathogenesis of myocardial hypertrophy
[90]	Murine model of lSO-induced HCM	NOS isoforms and ISO-induced SNO of Ca^2+^ regulating proteins in the heart
[91]	Murine I/R-IPC	IPC and GSNO-induced cardioprotection during MI
[71,92]	C203S-CypD knock-in MEF cell	The role of CypD SNO at cysteine 203 in activation of mPTP
[93]	C202S-CypD knock-in mouse	Investigation of PTMs at Cysteine 202 of CypD
[94]	GSNOR^−/−^ mice with IR	The role of GSNOR in the regulation of cardiovascular function in MI
[95]	GSNOR^−/−^ KO mice	Dynamic denitrosylation of GSNOR in myocardial performance
[96]	GSNOR KO mice	GSNOR in regeneration and proliferation of cardiomyocytes
[97]	GSNOR-Tg mice with ISO-HCM	GSNOR-mediated SNO of calcium-handling proteins in cross-talk with cardiac β-AR signaling in cardiac hypertrophy
[98]	GSNOR KO and Tg mice with TAC-HCM	GSNOR in regulation of hypertrophic cardiomyopathy
[99]	GSNOR-Tg mice	Cardioprotection of GSNO against sepsis-induced myocardial depression
[100,101]	Female murine I/R	Mechanism of ER-β-induced cardioprotection against MI
[102]	murine I/R model with pharmacologic IPC	Cardioprotection of adenosine A1 receptor agonist CHA in male and female hearts against MI
[103]	Human hearts	Ox-RAC and SNO-RAC in detection of protein oxidation and SNO levels in the left ventricle
[104]	GSNOR KO mice with ex vivo and in vivo I/R	Sex-dependent cardioprotection of GSNOR against MI
[105,106,107]	Dog IR model by LAD-ligation	Cardioprotection of inorganic sodium nitrite against I/R-induced severe ventricular arrhythmias

Legends for Table 2: I/R: ischemia/reperfusion; IPC: ischemic preconditioning; MI: myocardial ischemia; SNO: S-nitrosylation; SNO-MPG: S-nitroso-2-mercaptopropionyl glycine; Mito-SNO: mitochondria-targeted S-nitrosothiol; SNO-TRIM72: S-nitroso-tripartite motif-containing protein 72; SNO-GAPDH: S-nitrosylated glyceraldehyde-3-phosphate dehydrogenase; eNOS: endothelial nitric oxide synthase; NO: nitric oxide; SSM: subsarcolemmal mitochondria; mtCx43: mitochondrial connexin 43; sGC: soluble guanylyl cyclase; cGMP: cyclic guanosine monophosphate; PKG: protein kinase G; PostC: postconditioning; SNO-MLP: S-nitrosylation of the muscle LIM protein (LIM is the acronym of the first three family proteins identified: lin-11, islet-1 and mec-3); TAC: transverse aortic constriction; ISO: isoproterenol; HCM: hypertrophic cardiomyopathy; TLR3: toll-like receptor 3; CyD: Cyclophilin D; GSNO: S-nitrosoglutathione; MEF: mouse embryonic fibroblasts; mPTP: mitochondrial permeability transition pore; PTMs: post-translational modifications; GSNOR: S-nitrosoglutathione reductase; LCA: left coronary artery; GSNOR-Tg: GSNOR overexpression; β-AR: β-adrenergic receptor; ApoE: Apolipoprotein E; ERβ: estrogen receptor β; Ox-RAC: oxidation resin-assisted capture; CHA: N6-cyclohexyladenosine; LAD: left anterior descending artery.

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
