# Peer review of "Post-Translational S-Nitrosylation of Proteins in Regulating Cardiac Oxidative Stress"

_antioxidants, 2020, doi:10.3390/antiox9111051_

Round 1

Reviewer 1 Report

As requested, I reviewed the manuscript (ID antioxidants-961946) "Posttranslational S-nitrosylation of Proteins in Regulating Cardiac Oxidative Stress", by Xiaomeng Shi and Hongyu Qiu.

The proposed manuscript deals with the summarizing description of the cellular pathways and molecular mechanisms on which S-nitrosylation is based. The authors review various aspects, including the molecular basis and mechanisms of S-nitrosothiols (SNO) formation, the spatiotemporal specificity of SNO, and regulatory mechanisms of SNO homeostasis. Also, the current available methods for SNO detection are described. Finally, the topic of SNO in cardiac pathogenesis and protection has been addressed in a second major section.

The review manuscript is well conceived, with balance among the different main sections.

Overall, the discussed issues contain all the elements for comprehension of the subject, for understanding the cited results and for finding hints to deepen information.

The reading sounds like a good and comprehensible continuum, and there’s no feeling of fragmented information. The manuscript is quite conceived for giving the necessary keywords to enter the investigated subject, thus reaching this goal.

On this basis, the authors demonstrated scientific mastery of the research. My overall comment is positive about publishing the research article.

Nevertheless, I recommend a minor-but-fine revision of the English form throughout the entire document, for reaching a better final shape.

Thank you very much for your attention to my opinion.

Reviewer 2 Report

The manuscript presented is well redacted.  discussing in depth what is known about the molecular and biological basis of posttranslational S-nitrosylation of proteins. Particular interest was given to the involvement of this posttraslational process in the regulation of oxidative stress in several cardiac pathologies reporting the most recent studies conducted in different animal models and in human clinical trials. Interestingly, the current available methods applied for the detection of S-nitrosylation have been reported. I have no further request to suggest. In my opinion, this work can contribute significantly to the scientific field.

Author Response

Please the attachment

Reviewer 3 Report

The manuscript submitted for review is a very well planned, well-conducted and properly described review of the current state of knowledge in a selected current aspect of cardiac pathology.
In my opinion, its value can be increased by extending the topic to include the importance of the discussed mechanism in the pathology of blood vessels.
There is no discussion of studies on the importance of the analyzed pathomechanism in the pathology of arterial hypertension, atherosclerosis or stroke.
Due to the epidemiological importance, it would be worth discussing the significance of the analyzed mechanism in patients with type 2 diabetes.
The authors discuss the differences in the pathological significance of SNO between the sexes. However, there is no discussion about the importance of body weight or BMI for the discussed pathomechanism of cardiovascular diseases.

Round 2

Reviewer 3 Report

Thanks to the authors for making changes to their manuscript in line with my previous comments.I have no further comments.